# Application of the Extracts of *Uncaria tomentosa* in Endodontics and Oral Medicine: Scoping Review

**DOI:** 10.3390/jcm11175024

**Published:** 2022-08-26

**Authors:** Mario Dioguardi, Francesca Spirito, Diego Sovereto, Andrea Ballini, Mario Alovisi, Lorenzo Lo Muzio

**Affiliations:** 1Department of Clinical and Experimental Medicine, University of Foggia, Via Rovelli 50, 71122 Foggia, Italy; 2Department of Precision Medicine, University of Campania “Luigi Vanvitelli”, 80138 Naples, Italy; 3Department of Surgical Sciences, Dental School, University of Turin, 10127 Turin, Italy

**Keywords:** *Uncaria tomentosa*, endodontics, oral medicine, stomatitis, cat’s claw, *Enterococcus faecalis*

## Abstract

Background: The main purpose of endodontic treatment is to eliminate the bacteria that are responsible for the contamination and infection of the internal surfaces in order to resolve any pulp or periapical pathology. In fact, some bacteria, such as *Enterococcus faecalis*, can escape the action of root canal irrigants by aggregating into a biofilm and penetrating deeply into the dentinal tubules. *Uncaria tomentosa* is a plant belonging to the Rubiaceae family and also commonly known as cat’s claw due to the shape and position of the spines; it is a traditional Peruvian medicinal plant of Amazonian origin. Applications in the dental field have been described both in the prevention and treatment of stomatitis and as an antibacterial and anti-inflammatory agent; it has also been investigated as an additive in irrigants and specifically as gels in endodontic cements. The aim of this scoping review is to summarize all the scientific evidence on the possible applications of *Uncaria* tomentosa extracts in endodontics and, more generally, in oral medicine, in order to understand whether the active ingredients extracted from *Uncaria tomentosa* can bring a real advantage in endodontics, in the reduction of endodontic failures and in the onset of recurrent endodontic lesions. Methods: The scoping review was carried out strictly following the PRISMA-ScR checklist; the search was carried out on five databases (PubMed, Scopus, Science Direct, EBSCO and Web of Science) and a register (Cochrane library). Results: The research produced a number of bibliographic sources totaling 2104. With the removal of duplicates, 670 were obtained; potentially eligible articles amounted to 23, of which only seven in vitro studies (four microbiological studies), five clinical studies (three randomized trials) and a case report were included. Conclusions: From the data in the literature, it can be stated that the active ingredients present in *Uncaria tomentosa* could represent an interesting product to be used in the endodontic field, both in endocanalary cements and as a gel.

## 1. Introduction

The main purpose of endodontic treatment is to eliminate the bacteria that are responsible for the contamination and infection of the internal surfaces in order to resolve any pulp or periapical pathology. In addition, the treatment must aim at achieving an apical and coronal seal with the three-dimensional closure of the endocanalary gaps to prevent bacteria from penetrating into the endodontic space and colonizing the internal spaces not filled with root canal filling materials [1].

A fundamental role in obtaining a suitable seal and antimicrobial power is played by root canal sealants or, more generally, endodontic cements, which have antibacterial action, making the canal an unsuitable environment for bacterial proliferation. In fact, some bacteria, such as *Enterococcus faecalis*, can escape the action of root canal irrigants by aggregating into a biofilm and penetrating deeply into the dentinal tubules [2].

Sealing endodontic canals with materials that do not favor bacterial proliferation and that have bactericidal or proliferation-inhibiting power can be a winning strategy. Chlorhexidine (CHX) is one of the substances that has the ability to inhibit bacterial proliferation [3]. It is also used as a canal irrigant and is adsorbed on the dentinal surface, being slowly released later; in a similar way, other substances present in irrigants or endodontic cements can carry out this prolonged action over time, including many products of natural origin [4].

*Uncaria tomentosa* (Willd. ex Schult.) DC is a plant belonging to the *Rubiaceae* family and also commonly known as cat’s claw due to the shape and position of the spines; it is a traditional Peruvian medicinal plant of Amazonian origin [5], and has antioxidant [6], antimicrobial, antineoplastic, immunomodulating [7], antiretroviral [8] and anti-inflammatory properties. Its extracts contain oxindole [9], triterpenes, phenolic compounds [10], alkaloids [11], glycosides vegetable steroids, flavonoids and tannins, while the compound that has the greatest antibacterial properties, found in the bark, is isopteropodine-HCl, a pentacyclic oxindole alkaloid whose action is directed towards Gram-positive bacteria [12].

In the medical field, this plant has potential in the treatment of hyperglycemia, hyperlipidemia, metabolic syndrome and pregnancy hypertension [13,14], or as a dermo-cosmetic spray in patients with mild-to-moderate cutaneous pain [15], and it has also been tested as a binding factor for the ace2 receptor at the site where the SARS-CoV-2 spike protein binds [16,17,18]. It has demonstrated some anticancer properties, targeting breast cancer and melanoma cell lines [19].

Applications in the dental field have been described both in the prevention and treatment of stomatitis and as an antibacterial [20] and anti-inflammatory [21] agent; it has also been investigated as an additive in irrigants, and specifically as gels and in endodontic cements, to exploit the ability of the active ingredients to be adsorbed by the tooth surface with a mechanism similar to CHX and to be released [22], successively slowly exerting its inhibitory action on bacterial proliferation.

There are no systematic reviews or scoping reviews on the role *of Uncaria tomentosa* extracts in use in endodontics. The aim of this review is to summarize all the scientific evidence on the possible applications of *Uncaria tomentosa* extracts in endodontics and, more generally, in oral medicine in order to understand whether the active ingredients extracted from *Uncaria tomentosa* can bring a real advantage in endodontics, in the reduction of endodontic failures and in the onset of recurrent endodontic lesions.

## 2. Materials and Methods

### 2.1. Protocol and Registration

The scoping review was carried out strictly following the PRISMA-ScR checklist (PRISMA Extension for Scoping Reviews), as described by Tricco et al. [23] (the complete checklist is available in the Appendix A). The scoping review protocol was registered prior to its execution on the International Platform of Registered Systematic Review and Meta-Analysis Protocols (INPLASY), with registration number INPLASY 202270024 and DOI number 10.37766/inplasy2022.7.0024.

### 2.2. Eligibility Criteria

All studies that addressed *Uncaria tomentosa* in association with endodontic pathologies and, more generally, with oral pathologies were considered potentially eligible, and no restrictions were applied in relation to the year of publication or according to the language, provided that an abstract was available in the English, Spanish or Portuguese languages (the choice of Spanish and Portuguese in association with English was due to the fact that, since *Uncaria tomentosa* is a traditional medical plant of the Amazon region, excluding articles in this language would have potentially led to publication bias and to failure to retrieve reports in gray literature sources); literature reviews were excluded and were only used as bibliographic research sources.

### 2.3. Information Sources

The search was carried out on 5 databases (PubMed, Scopus, Science Direct, EBSCO and Web of Science) and a register (Cochrane Library); in addition, a gray literature search was performed on Google Scholar and Opengray (DANS EASY Archive). Potentially eligible articles were also searched among references from literature reviews on *Uncaria tomentosa*.

The research was conducted between 15 June 2022 and 1 July 2022, with a final update of the records identified on 3 July 2022.

### 2.4. Search

The authors responsible for researching the studies used the following keywords in the databases: *Uncaria tomentosa* OR cat’s claw. The key words used on PubMed are shown below: Search: uncaria tomentosa OR cat’s claw, Sort by: Most Recent “cat s claw”[MeSH Terms] OR (“cat s”[All Fields] AND “claw”[All Fields]) OR “cat s claw”[All Fields] OR (“uncaria”[All Fields] AND “tomentosa”[All Fields]) OR “uncaria tomentosa”[All Fields] OR (“cat s claw”[MeSH Terms] OR (“cat s”[All Fields] AND “claw”[All Fields]) OR “cat s claw”[All Fields]).Translations uncaria tomentosa: “cat’s claw”[MeSH Terms] OR (“cat’s”[All Fields] AND “claw”[All Fields]) OR “cat’s claw”[All Fields] OR (“uncaria”[All Fields] AND “tomentosa”[All Fields]) OR “uncaria tomentosa”[All Fields] cat’s claw: “cat’s claw”[MeSH Terms] OR (“cat’s”[All Fields] AND “claw”[All Fields]) OR “cat’s claw”[All Fields].

### 2.5. Selection of Sources of Evidence

The search for potentially eligible studies was conducted by 2 reviewers (M.D. and D.S.), with a third reviewer (A.B.) given the task of choosing whether to include the studies in situations of conflict.

The 2 reviewers, after having decided jointly on the eligibility criteria, the databases to be used and the keywords, independently carried out the research work, reporting the number of records obtained for each keyword and each database used. Duplicate records from different databases were removed using EndNote 9 software, and study overlays that could not be uploaded to EndNote were manually removed after the screening phase. Always independently, they proceeded to the screening and inclusion of the studies from the records obtained, and only subsequently was there a comparison of the included studies between the 2 reviewers.

### 2.6. Data Charting Process, Data Items, Synthesis of Results and Critical Appraisal of Individual Sources of Evidence (Risk of Bias)

The type of data to be extracted was decided in advance by the authors; the data to be extracted concerned the lead author, the date, the country in which the study was performed, the type of bacteria used, the compounds or materials tested and the main results. The data extracted from the studies were reported in word tables independently by the 2 reviewers and subsequently compared. The data obtained were then represented through tables and are included in the Results section of this manuscript, A risk of bias assessment was also conducted for both in vitro studies and clinical trials (performing a risk of bias assessment is generally more appropriate for systematic reviews of the literature, but it is also useful to perform within a scoping review).

## 3. Results

### 3.1. Selection of Sources of Evidence

The research in Science Direct, SCOPUS, EBSCO, Web of Science, PubMed and the Cochrane Library produced a number of bibliographic sources equal to 2104. With the removal of duplicates, 670 were obtained; potentially eligible articles totaled 23, of which only 11 fully complied with the criteria of eligibility.

In addition, the gray literature analysis (http://www.opengrey.eu, accessed on 1 July 2022, DANS EASY Archive and Google Scholar) and previous systematic reviews allowed for the identification of two additional studies to be included in the quantitative assessment (Figure 1). The entire procedure for the identification, selection and inclusion of the studies is indicated in the flowchart in Figure 1.

### 3.2. Characteristics of Sources of Evidence and Results of Individual Sources of Evidence

A total of 13 articles were included in the scoping review:
✓three studies for the primary outcome—studies describing a possible application of the tomentose claw extracts in endodontics: Herrera et al., 2016 [24], Herrera et al., 2010 [25], Caldas et al., 2021 [22];✓10 studies for the secondary outcome—studies that concerned the possible application of *Uncaria tomentosa* in medicine and oral pathology: Tay et al., 2015 [26], Tay et al., 2014 [21], Paiva et al., 2009 [27], Silva et al., 2021 [28], Vergiú 2006 [29], Caldas et al., 2010 [30].

In total, seven in vitro studies (four microbiological studies), five clinical studies (three randomized trials) and a case report were included. Specifically, a total of 188 patients were recruited in clinical trials (170 excluding patients who did not continue the follow-up).

In the clinical studies, the main oral pathology considered was aerating Candida stomatitis (three studies), followed by one study on cold sores and the accumulation of tartar typology. Of the four in vitro microbiological studies, the main bacteria investigated were *E. facaelis* (three studies), *C. albicans* (two studies) and *S. aureus* (two studies). All extracted data are reported in Table 1, Table 2 and Table 3.

### 3.3. Risk of Bias

The risk of bias was assessed based on the Checklist for Reporting In Vitro Studies (CRIS) guidelines, proposed to evaluate in vitro dental studies [34]. The results are shown in Table 4; the score for each category is assigned in a range from 1 to 3.

In the risk of bias assessment for in vitro studies, the studies that presented high quality were those of Herrera et al. [24] and Polassi et al. [31]; furthermore, they were the only studies describing the blinded randomization of the samples.

Meanwhile, for the randomized clinical trials, a risk of bias assessment was performed using the points described in the Cochrane Handbook, Chapter 8 (Assessing risk of bias in included studies).

The studies were evaluated using three parameters: low risk of bias, high risk of bias, unclear. The graphs for the calculation of the risk of bias were produced using the software ReV Manager 5.4 (Cochrane Collaboration, Copenhagen, Denmark) (Figure 2); the case report was excluded from the assessment of the risk of bias [26].

## 4. Discussion

### 4.1. Summary of Evidence

The authors performed a scoping review regarding the potential application of *Uncaria tomentosa* in endodontics and oral medicine; this review represents the first review performed specifically on *Uncaria tomentosa* in the field of oral pathology.

From the analysis of the data present in the literature, it is clear that extracts of *Uncaria tormentosa* can find application in endodontics and in medicine and oral pathology; in endodontics, its possible application includes both in endodontic cements thanks to a slow-release antibacterial effect against *Enterococcus faecalis* and other bacteria, and as a gel or liquid in addition to other endodontic irrigants. Furthermore, the analysis of the physical properties would make it suitable for use in formulations for background materials, and as a whitening agent.

In medicine and oral diseases, the main field of application is undoubtedly gels or mouthwashes in the treatment of Candida stomatitis and as a mild anti-inflammatory. The main effects demonstrated can therefore be summarized as follows: antimicrobial effects, aimed at C. albicans and *Enterococcus faecalis*, and anti-inflammatory and whitening and anti-tartar effects (Table 1, Table 2 and Table 3).

This scoping review included 13 studies, including five clinical studies, seven in vitro studies and a case report. In total, 170 patients were included in follow-up treatments and the main pathology investigated was oral Candida stomatitis.

In the endodontic field, three in vitro studies have investigated the main uses of *Uncaria tomentosa* (UT) extracts—specifically, two studies considered it as a gel to be used in the cleaning and shaping phase of endodontic canals, and one study as a filler, used in addition to a cement as a sealant having slow-release antimicrobial activity in root canal closure.

In particular, Herrera et al. (2016) [24] noted a similar bacterial load reduction for 2% CHX, 2% UT gel and 2% sodium hypochlorite on *E. feacalis*-contaminated dentin (ATCC 29212). These data confirm the results of a previous in vitro study conducted by the same research group, Herrera et al., 2010 [25], but which involved various microbiological agents that may be responsible for the persistence of endodontic lesions: *E. faecalis* (ATCC 29212), *S. aureus* (ATCC 95106) and *C. albicans* (ATCC 10231). This last study reports a synergistic effect when UT and CHX extracts are present in the same gel formulation.

Caldas et al., 2021 [22] instead reported that the bacterial load is further reduced when MTA and AH plus are mixed with herbal remedies derived from UT. These three in vitro microbiological studies, in association with the salivary cultural data of 100 patients reported in the study by Ccahuana-Vasquez et al., 2007 [20], suggest an antibacterial action that UT directs towards many bacteria involved in the persistence of endodontic lesions (*E. faecalis*), although the data of Vasquez et al. do not show an inhibitory action for *C. albicans* and *P. aeruginosa* in the culture.

The other three in vitro studies evaluated the effects of UT extracts on various aspects that may be of interest in conservative dentistry. Polassi et al., 2021 tested the effects of UT on dentin by investigating various aspects related to adhesion and specifically to the strength of the adhesive bond. The UT in this study did not cause a reduction in the adhesive bond but had a negative effect on the elasticity of the dentin, determined in part by a demineralizing action [31]. de la Fuente et al. tested UT bark powder by combining it with polymethacrylate silica oxide and calcium oxide as a potential filling material, finding a pH of 6.8 and physical and chemical characteristics that would make it suitable as a filling substrate material [33], while Garcia et al., 2008 investigated the plant’s antioxidant characteristics as a whitening agent for enamel and dentin [32].

In the literature, there are five clinical studies performed on *Uncaria tomentosa* and diseases closely related to oral medicine and pathology; three studies, including two randomized control trials, investigated the effects of UT on stomatitis whose main etiological agent was *C. albicans*.

Paiva et al., 2009 [27] presented the results of a cross-sectional observational study that was the first published clinical study on UT and stomatitis. Pavia et al. tested both the topical application of UT gel and the topical application of 4% miconazole on 20 patients with Candida stomatitis, diagnosed with microbiological culture tests (30 recruited), and the results were superior for the miconazole group compared to the UT gel extract group; the data shown by Pavia et al. indicate that although some in vitro tests showed antimicrobial activity against Candida pathogens (Ccahuana-Vasquez et al., 2007 [20] did not report any inhibitory effect against *C. albicans* in their in vitro study, unlike Herrera et al., 2010 [25]), applying this information at a clinical level, the UT extracts showed lower performance than miconazole, with an advantage only in terms of adverse reactions for UT.

Tay et al., 2014 [21] conducted a randomized control trial in which they recruited 50 patients and compared the topical application of 2% miconazole, placebo or 2% UT gel in patients with Candida stomatitis. The data reported by Tay et al. indicate efficacy of the 2% UT gel similar to 2% miconazole rather than the placebo in the reduction of symptoms after 7–14 days. The reasons for the comparable effect in this study may have resulted from the confounding factors reported by the authors; in fact, patients who were prosthesis wearers were given correct information on maintaining the hygiene of the prosthesis in order to reduce the microbiological load on these surfaces, so this may have led to a regression of Candida symptoms in the three groups in a similar way. In principle, these data are in partial agreement with those of Pavia et al., 2009 [27] and Silva et al., 2021 [28].

Silva et al. [28] compared the following compounds—nystatin, UT, propolis and placebo—in 38 patients with Candida stomatitis; the results are in line with those of Tay et al., 2014, demonstrating an antifungal effect.

An antiviral effect of UT extracts against Herpes viruses (HSV1) was tested by comparing it with acyclovir in a randomized double-blind trial conducted by Caldas et al., 2010 [30], on 31 patients with 51 overall episodes of Herpes labialis. This study showed the plant’s greater efficacy as an anti-inflammatory, while there was no difference in terms of the duration of the herpetic pathology, as well as the formation of crusty lesions typical of Herpes labialis. It should also be noted that the topical application of UT as a product with functions similar to mouthwashes reduced the formation of tartar (in 15–45 days) in a randomized study conducted by Vergiú 2006, presented as a doctoral thesis work at the Universitat Internacional de Catalunya [29].

The data presented in the literature certainly demonstrate antibacterial and antifungal activity due to the presence of bioactive principles; it is also worth noting the anti-inflammatory capacity. These characteristics make UT extracts a suitable candidate in applications in endodontic use, because the antimicrobial activity demonstrated against *Enterocccus faecalis* is also expressed after the application of the endodontic gel, since the compound is partially adsorbed by the dentinal surface, and the antimicrobial effect is also expressed when the compound is present in formulations with other endodontic cements [22,24,25].

Furthermore, the anti-inflammatory capacity shown can reduce the inflammatory symptoms linked to endodontic pathology and endodontic treatment. The literature is also in agreement regarding its use as an adjuvant for the treatment of Candida stomatitis, with promising results [35].

### 4.2. Limitations

The main limitations of this scoping review are to be found in the non-prolific scientific literature on the subject; an attempt was made to minimize publication bias by searching for all the reports in the gray literature and not imposing language limitations. In fact, manuscripts and reports in Portuguese and Spanish, or in other languages containing at least one abstract in English, were considered [36,37].

## 5. Conclusions

From the data in the literature, it can be stated that the active ingredients present in *Uncaria tormentosa* could represent an interesting product to be used in the endodontic field, both in endocanal cements and as a gel to be applied during the shaping and cleansing phase, thanks to the antimicrobial and anti-inflammatory properties.

Furthermore, the results obtained from the randomized studies and trials that have shown action against Candida stomatitis, with similar effects in some studies to antifungals (miconazole and nicostatin), are interesting.

## Figures and Tables

**Figure 1 jcm-11-05024-f001:**
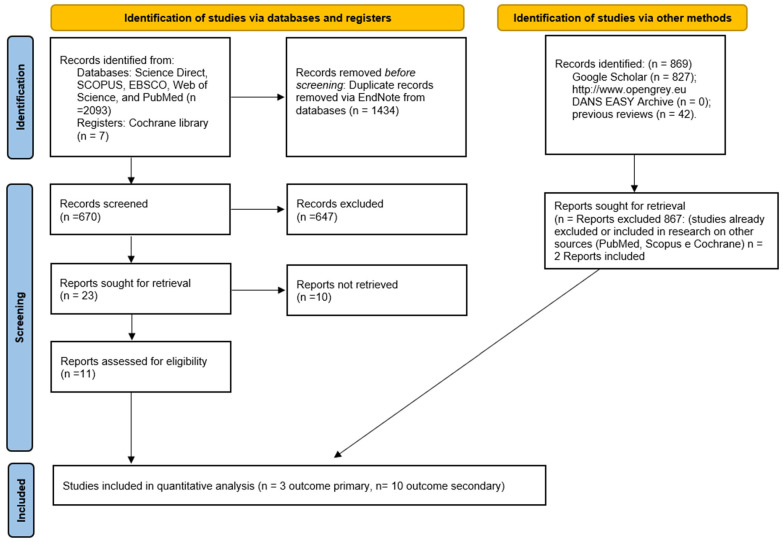
Entire selection and screening procedures are described in the PRISMA flowchart.

**Figure 2 jcm-11-05024-f002:**
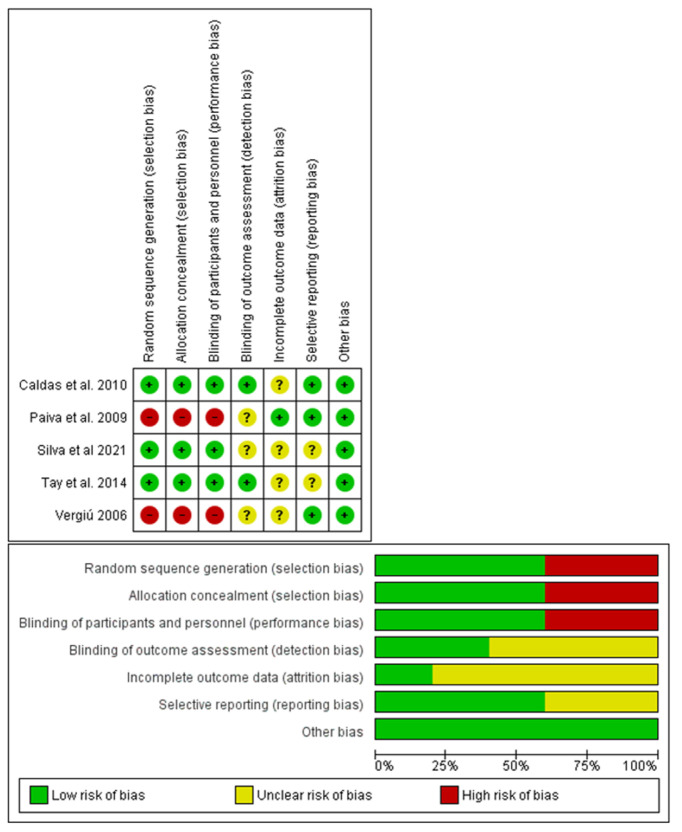
Risk of bias graph for each of the studies included in this review; Tay et al., 2014 [21], Paiva et al., 2009 [27], Silva et al., 2021 [28], Vergiú 2006 [29], Caldas et al., 2010 [30].

**Table 1 jcm-11-05024-t001:** In vitro microbiological studies, UT (*Uncaria tomentosa*).

Study Characteristics						Study Results
First Author, Date	Country	Type of Study	Bacteria	Sample	Irrigant or Sealer	Main Findings of the Study
Herrera et al., 2016 [24]	Brazil	In vitro	*E. faecalis* (ATCC 29212)	Culture (infected root canal dentin)	UT, CHX, NaOCl	Antibacterial effect of 2% UT gel and 2% CHX against *E. faecalis* in infected root canal dentin.
Herrera et al., 2010 [25]	Brazil	In vitro	*E. faecalis* (ATCC 29212), *Staphylococcus aureus* (ATCC 95106), *C. albicans* (ATCC 10231)	Culture	CHX, UT, CHX + UT	CHX + UT was the substance most effective against *E. faecalis* and *C. albicans.*
Caldas et al., 2021 [22]	Brazil	In vitro	*E. faecalis* (ATCC 29212)	Culture	AH Plus (Control—0% UT), AH Plus 2% (AH Plus + 2% UT), AH Plus 5% (AH Plus + 5% UT), MTA Fillapex (Control—0% UT), MTA Fillapex 2% (MTA Fillapex + 2% UT) and MTA Fillapex 5% (MTA Fillapex + 5% UT)	The incorporation of phytotherapic UT decreased cytotoxicity and raised the antimicrobial action of root canal sealers.
Ccahuana-Vasquez et al., 2007 [20]	Brazil	In vitro	*S. mutans*, *S. aureus* *S. intermedius*, *C. albicans*, *K. pneumoniae*, *K. oxytoca*, *K. Terrigena*, *E. cloacae*, *E. sakazakii*, *E. asburiae*, *E. amnigenus*, *E. coli*, *C. freundii*, *C. amalonaticus* *S. liquefaciens*, *S. odorifera*, *Pantoea* spp., *P. aeruginosa*	Culture (one hundred and six strains from the human oral cavity)	0.25%, 0.5%, 1%, 2%, 3%, 4%, 5% of UT	UT presented antimicrobial activity on *Enterobacteriaceae*, *S. mutans* and *Staphylococcus* spp. *isolates*; however, it did not present inhibitory effect on *P. aeruginosa* and *C. albicans.*

**Table 2 jcm-11-05024-t002:** Case reports and clinical studies.

Study Characteristics						Study Results
First Author, Date	Country	Type of Study	Pathologies	Patients	Treatment Groups	Main Findings of the Study
Tay et al., 2015 [26]	Perú, Brazil	Case report	Stomatitis caused by *C. albicans*	1	2% UT gel against denture stomatitis	UT gel was an effective topical adjuvant treatment in oral candidiasis.
Tay et al., 2014 [21]	Brazil	Randomized clinical study	Stomatitis caused by Candida ^a^	50 (5 male, 43 female) *	2% miconazole, placebo, 2% UT in gel	UT gel was an effective topical adjuvant treatment in oral candidiasis.
Paiva et al., 2009 [27]	Brazil	Clinical observational cross-sectional study	Stomatitis caused by Candida ^c^	30 ^b^	4% miconazole, 10 grams UT in gel	Therapeutic efficacy of Miconazole slightly higher than UT.
Silva et al., 2021 [28]	Brazil	Randomized clinical study	Stomatitis caused by Candida	37 (28 female, 2 male)	Sterile distilled water, Nystatin oral suspension, 20% alcoholic extract propolis, Punica granatum Linné gel and UT gel	*Punica granatum* L. gel and UT gel adjuvant treatment for stomatitis.
Vergiú 2006 [29]	Spain	Clinical study	Accumulation of Tartarus	40 patients	20 with consumption of UT tincture, 20 controls	Significant difference in the accumulation of plaque and tartar after 15, 30 and 45 days.
Caldas et al., 2010 [30]	Brazil	Randomized clinical study	Herpes Labialis	31 patient (51 Herpes episodes)	27 UT, 27 Zovirax	UT had better efficacy only as an anti-inflammatory.

* 2 patients withdrawn, ^a^ *C. Albicans C. tropicalis C. glabrata* and *C. krusei*, ^b^ only 20 patients were positive for Candida on mycological analysis, ^c^
*C. albicans C.*, *tropicalis* and *C. guilliermondi*.

**Table 3 jcm-11-05024-t003:** Non-microbiological in vitro studies.

Study Characteristics			Study Results
Author, Date	Country	Substances Tested or Origin of the Main Substances Tested	Main Findings of the Study
Polassi et al., 2021 [31]	Brazil	*Dysphania ambrosioides*, UT, *Paullinia cupana*, *Rhus chinensis*, *Acacia decurrens*	Bioactive plant extracts among which UT do not affect the strength of adhesion to dentin, but a negative influence on the form is reported for elasticity of demineralized dentin.
Garcia et al., 2008 [32]	Brazil	10% ascorbic acid solution, 10% ascorbic acid gel, 10% sodium ascorbate solution, 10% sodium ascorbate gel, 10% sodium bicarbonate, Neutralize^®^, Desensibilize^®^, catalase C-40 at 10 mg/mL, 10% alcohol solution of alpha-tocopherol, Listerine^®^ (LIS), 0.12% CH, Croton Lechleri, 10% aqueous solution UT, artificial saliva and 0.05% sodium fluoride	Tested the potential of plant extracts including UT; UT is antioxidant and has translational potential as teeth whitening agent
de la Fuente et al. [33]	Brazil ^1^	UT	Evaluation of the pH (6.81) of UT extracts and evaluation of its handling and resistance characteristics as a material in use in conservation in association with polymeticrylate silica oxide and calcium oxide.

^1^ it is not clearly specified whether the country in which the study was conducted was Brazil.

**Table 4 jcm-11-05024-t004:** Assessment of the risk of bias within the studies, with scores of 5 to 9 indicating low quality, 10 to 12 intermediate quality and 13 to 15 high quality.

First Author, Date	Sample Size Calculation	Meaningful Difference between Groups	Sample Preparation and Handling	Allocation Sequence, Randomization and Blinding	Statistical Analysis	Score
Herrera et al., 2016	3	3	3	3	2	14
Herrera et al., 2010 [25]	3	3	2	2	1	11
Caldas et al., 2021 [22]	2	3	3	1	3	12
Ccahuana-Vasquez et al., 2007 [20]	2	3	2	1	2	10
Polassi et al., 2021 [31]	3	3	3	2	3	14
Garcia et al.2008 [32]	3	3	3	1	2	12
de la Fuente et al. [33]	3	3	3	1	2	12

## Data Availability

Not applicable.

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
