# Peer review of "Application of the Extracts of Uncaria tomentosa in Endodontics and Oral Medicine: Scoping Review"

_jcm, 2022, doi:10.3390/jcm11175024_

Round 1
Reviewer 1 Report
Review “Application of the Extracts of Uncaria Tomentosa in Endodontics and Oral Medicine: Scoping Review” by Mario Dioguardi et al 2022
The authors of “Application of the Extracts of Uncaria Tomentosa in Endodontics and Oral Medicine: Scoping Review” present a bibliographic review of the application of Extracts of Uncaria in oral medicine. The review was well structured and done and effort to perform a non-bayous analysis. However, in discussion and the tables is not enough information to allow the reader to understand what the magnitude of the positive effects of the Extracts of Uncaria Tomentosa would be. In addition, there were some mistakes in tables as described below:
Results:
In table 1 there are some mistakes that need to be reviewed:
· Second row, last column; description is in Italian “e CHX+UT era la sostanza più efficace contro E. faecalis e C. albic”
· Third row, last column, missing the T in The “he incorporation of phytotherapic UT decreased cytotoxicity, and raised the antimicrobial action of root canal sealers”
· Fourth row, fifth column; “… hu-man oral Cavity…”
In table 2 there are some mistakes that need to be reviewed:
· Fourth row, Last two columns are in Italian “ 4% miconazole, 10 grammi UT in gel.” And “Efficacia terapeutica del Miconazolo leggermente superiore a UT”
In table 3:
First row, last column , is missing the capital letter of the first word; “bioactive plant extracts”
Last row, second column, “Brazil?” introduction of interrogation sign…”
Discussion:
Authors mention “ These 3 in vitro microbiological studies ---” In vitro should be in italic
Authors mention “ non 228 riporta nel suo studio in vitro alcun effetto inibitorio contro C.albicans, a differenza di 229 Herrera et al. 2010 [24])” did not translate to English…
The discussion would benefit of the introduction of numbers of the positive effects of Extracts of Uncaria Tomentosa, discussion did not provide enough details in results presented by the studies chosen for an evaluation of the magnitude of the treatments with Extracts of Uncaria Tomentosa.
Author Response
Reviewer 1
Review “Application of the Extracts of Uncaria Tomentosa in Endodontics and Oral Medicine: Scoping Review” by Mario Dioguardi et al 2022
The authors of “Application of the Extracts of Uncaria Tomentosa in Endodontics and Oral Medicine: Scoping Review” present a bibliographic review of the application of Extracts of Uncaria in oral medicine. The review was well structured and done and effort to perform a non-bayous analysis. However, in discussion and the tables is not enough information to allow the reader to understand what the magnitude of the positive effects of the Extracts of Uncaria Tomentosa would be. In addition, there were some mistakes in tables as described below:
Results:
In table 1 there are some mistakes that need to be reviewed:
- Second row, last column; description is in Italian “e CHX+UT era la sostanza più efficace contro E. faecalis e C. albic”
- Third row, last column, missing the T in The “he incorporation of phytotherapic UT decreased cytotoxicity, and raised the antimicrobial action of root canal sealers”
- Fourth row, fifth column; “… hu-man oral Cavity…”
In table 2 there are some mistakes that need to be reviewed:
- Fourth row, Last two columns are in Italian “ 4% miconazole, 10 grammi UT in gel.” And “Efficacia terapeutica del Miconazolo leggermente superiore a UT”
In table 3:
First row, last column , is missing the capital letter of the first word; “bioactive plant extracts”
Last row, second column, “Brazil?” introduction of interrogation sign…”
Discussion:
Authors mention “ These 3 in vitro microbiological studies ---” In vitro should be in italic
Authors mention “ non 228 riporta nel suo studio in vitro alcun effetto inibitorio contro C.albicans, a differenza di 229 Herrera et al. 2010 [24])” did not translate to English…
The discussion would benefit of the introduction of numbers of the positive effects of Extracts of Uncaria Tomentosa, discussion did not provide enough details in results presented by the studies chosen for an evaluation of the magnitude of the treatments with Extracts of Uncaria Tomentos
ANSWER
Thank you for your comments and suggestions and for taking the time to review the manuscript. His advice was useful in order to improve in the discussion of the possible effects and uses of the uncaria tomentosa
All errors in the text have been corrected as suggested including untranslated parts.
moreover, as suggested to me, the following part concerning the uses and main effects of the tormenting uncaria has been added:
From the analysis of the data present in the literature, it is clear that the areas of application of the extracts of Uncaria tormentosa can find application in endodontics and in medicine and oral pathology; in endodontics, its possible application can be both in use in endodontic cements thanks to a slow-release antibacterial effect against enterococcus faecalis and other bacteria and as a gel or liquid in addition to other endodontic irrigants. Furthermore, the analysis of the physical properties would make it suitable for use in formulations for background materials, and as a whitening agent.
In medicine and oral diseases, the main field of application is undoubtedly the name gel or mouthwash in the treatment of candida stomatitis and as a mild anti-inflammatory. The main effects demonstrated can therefore be summarized as: antimicrobials aimed at C albicans and enterococcus faecalis, in anti-inflammatory and whitening and an-ti-tartar effects (table 1-3).

Reviewer 2 Report
Please rewrite the whole text following PRISMA guidelines.
Moreover split the data in the tables into study charactersitics, study results.
Please also add the quality and risk of bias assessment for all studies (which should be different according to the different study designs).
Author Response
Please rewrite the whole text following PRISMA guidelines.
Moreover split the data in the tables into study charactersitics, study results.
Please also add the quality and risk of bias assessment for all studies (which should be different according to the different study designs).
Answer
Thanks for the comments and suggestions given.
Your advice was helpful in improving the manuscript and implementing the risk of bias assessment in the review.
- Please rewrite the whole text following PRISMA guidelines.
The manuscript has been revised for the implementation of the non-discussed points present in the PRISMA-ScR, moreover a check list has been produced inserted in the supplementary files where it is reported where all the items of the PRISMA-ScR are present (page and line),the check list is also present at the bottom of this document.
- Moreover split the data in the tables into study charactersitics, study results.
The tables have been split as required by adding: study charactersitics, study results.
- Please also add the quality and risk of bias assessment for all studies (which should be different according to the different study designs).
The risk of bias was added as required for in vitro studies and clinical trials:
3.3. Risk of Bias
The risk of bias was assessed based on the Checklist for Reporting In vitro Studies (CRIS) guidelines ,proposed to evaluate in vitro dental studies [34]. The results are shown in Table 4; the score for each category is assigned with a range from 1 to 3.
Table 4. Assessment of the risk of bias within the studies, with scores 5 to 9 low quality, 10 to 12 intermediate quality, and 13 to 15 high quality.
First Author, Data |
Sample size calculation |
Meaningful difference between groups |
Sample preparation and handling |
Allocation sequence, randomization and blinding |
Statistical analysis |
Score |
Herrera et al. 2016 |
3 |
3 |
3 |
3 |
2 |
14 |
Herrera et al. 2010 [25] |
3 |
3 |
2 |
2 |
1 |
11 |
Caldas et al. 2021[22] |
2 |
3 |
3 |
1 |
3 |
12 |
Ccahuana-Vasquez et al. 2007 [20] |
2 |
3 |
2 |
1 |
2 |
10 |
Polassi et al. 2021 [31] |
3 |
3 |
3 |
2 |
3 |
14 |
Garcia et al.2008 [32] |
3 |
3 |
3 |
1 |
2 |
12 |
de la Fuente et al. [33] |
3 |
3 |
3 |
1 |
2 |
12 |
In the risk of bias assessment for in vitro studies, the studies that present a high quality are those of Herrera et al. [24] and Polassi et al. [31] furthermore, they are the only studies describing the blinded randomization of the samples.
While for the randomized clinical trials a risk of bias assessment was performed using the points described in the Cochrane Handbook chapter 8 (Assessing risk of bias in included studies)
The studies were evaluated using 3 parameters: Low risk of bias, High risk of bias, Unclear. The graphs for the calculation of the Risk of bias were performed using the software ReV Manager 5.4 (Cochrane Collaboration, Copenhagen, Denmark) (Figures 2), the case report was excluded from the assessment of the risk of bias [26].
Figure 2. Risk of bias graph for each of the studies included in this review.
Preferred Reporting Items for Systematic reviews and Meta-Analyses extension for Scoping Reviews (PRISMA-ScR) Checklist
SECTION |
ITEM |
PRISMA-ScR CHECKLIST ITEM |
REPORTED ON PAGE # |
TITLE |
|||
Title |
1 |
Identify the report as a scoping review. |
Page 1, line 1 |
ABSTRACT |
|||
Structured summary |
2 |
Provide a structured summary that includes (as applicable): background, objectives, eligibility criteria, sources of evidence, charting methods, results, and conclusions that relate to the review questions and objectives. |
Page 1, line 14-34 |
INTRODUCTION |
|||
Rationale |
3 |
Describe the rationale for the review in the context of what is already known. Explain why the review questions/objectives lend themselves to a scoping review approach. |
Page 2-3, line 44-77 |
Objectives |
4 |
Provide an explicit statement of the questions and objectives being addressed with reference to their key elements (e.g., population or participants, concepts, and context) or other relevant key elements used to conceptualize the review questions and/or objectives. |
Page 3, line 78-83 |
METHODS |
|||
Protocol and registration |
5 |
Indicate whether a review protocol exists; state if and where it can be accessed (e.g., a Web address); and if available, provide registration information, including the registration number. |
Page 2, line 85 90 |
Eligibility criteria |
6 |
Specify characteristics of the sources of evidence used as eligibility criteria (e.g., years considered, language, and publication status), and provide a rationale. |
Page 2-3, line 91- 101 |
Information sources* |
7 |
Describe all information sources in the search (e.g., databases with dates of coverage and contact with authors to identify additional sources), as well as the date the most recent search was executed. |
Page 3, line 103-109 |
Search |
8 |
Present the full electronic search strategy for at least 1 database, including any limits used, such that it could be repeated. |
Page 3, line 112-113 |
Selection of sources of evidence† |
9 |
State the process for selecting sources of evidence (i.e., screening and eligibility) included in the scoping review. |
Page 3, line 123-135 |
Data charting process‡ |
10 |
Describe the methods of charting data from the included sources of evidence (e.g., calibrated forms or forms that have been tested by the team before their use, and whether data charting was done independently or in duplicate) and any processes for obtaining and confirming data from investigators. |
Page 3, line 136- 142 |
Data items |
11 |
List and define all variables for which data were sought and any assumptions and simplifications made. |
Page 3, line 136- 142 |
Critical appraisal of individual sources of evidence§ |
12 |
If done, provide a rationale for conducting a critical appraisal of included sources of evidence; describe the methods used and how this information was used in any data synthesis (if appropriate). |
Page 3 line (142-146) |
Synthesis of results |
13 |
Describe the methods of handling and summarizing the data that were charted. |
Page 3, line 136- 142 |
RESULTS |
|||
Selection of sources of evidence |
14 |
Give numbers of sources of evidence screened, assessed for eligibility, and included in the review, with reasons for exclusions at each stage, ideally using a flow diagram. |
Page 4, line 148-160, figure 1. |
Characteristics of sources of evidence |
15 |
For each source of evidence, present characteristics for which data were charted and provide the citations. |
Page 4-5, line 161-179 |
Critical appraisal within sources of evidence |
16 |
If done, present data on critical appraisal of included sources of evidence (see item 12). |
Page 7-8, line 187-207 |
Results of individual sources of evidence |
17 |
For each included source of evidence, present the relevant data that were charted that relate to the review questions and objectives. |
Table 1,2,3 |
Synthesis of results |
18 |
Summarize and/or present the charting results as they relate to the review questions and objectives. |
Page 5, line 170-178 |
DISCUSSION |
|||
Summary of evidence |
19 |
Summarize the main results (including an overview of concepts, themes, and types of evidence available), link to the review questions and objectives, and consider the relevance to key groups. |
Page 8-10 |
Limitations |
20 |
Discuss the limitations of the scoping review process. |
Page 10, line 295-299 |
Conclusions |
21 |
Provide a general interpretation of the results with respect to the review questions and objectives, as well as potential implications and/or next steps. |
Page 10, line 30-307 |
FUNDING |
|||
Funding |
22 |
Describe sources of funding for the included sources of evidence, as well as sources of funding for the scoping review. Describe the role of the funders of the scoping review. |
Page 10, line 312 |
JBI = Joanna Briggs Institute; PRISMA-ScR = Preferred Reporting Items for Systematic reviews and Meta-Analyses extension for Scoping Reviews.
* Where sources of evidence (see second footnote) are compiled from, such as bibliographic databases, social media platforms, and Web sites.
† A more inclusive/heterogeneous term used to account for the different types of evidence or data sources (e.g., quantitative and/or qualitative research, expert opinion, and policy documents) that may be eligible in a scoping review as opposed to only studies. This is not to be confused with information sources (see first footnote).
‡ The frameworks by Arksey and O’Malley (6) and Levac and colleagues (7) and the JBI guidance (4, 5) refer to the process of data extraction in a scoping review as data charting.
- The process of systematically examining research evidence to assess its validity, results, and relevance before using it to inform a decision. This term is used for items 12 and 19 instead of "risk of bias" (which is more applicable to systematic reviews of interventions) to include and acknowledge the various sources of evidence that may be used in a scoping review (e.g., quantitative and/or qualitative research, expert opinion, and policy document).
From: Tricco AC, Lillie E, Zarin W, O'Brien KK, Colquhoun H, Levac D, et al. PRISMA Extension for Scoping Reviews (PRISMAScR): Checklist and Explanation. Ann Intern Med. 2018;169:467–473. doi: 10.7326/M18-0850.

Round 2
Reviewer 1 Report
The authors have corrected and included the information required by the reviewers
Reviewer 2 Report
Thank you for making the requested changes to the Manuscript.